# An atlas of late prenatal human neurodevelopment resolved by single-nucleus transcriptomics

Susana I. Ramos[1,2,6], Zarmeen M. Mussa[1,2,6], Elisa N. Falk [2,3], Balagopal Pai [1,2], Bruno Giotti[4], Kimaada Allette[4], Peiwen Cai [4], Fumiko Dekio[1], Robert Sebra[4,5], Kristin G. Beaumont [4], Alexander M. Tsankov [4,5] ✉ & Nadejda M. Tsankova [1,2,5] ✉

Late prenatal development of the human neocortex encompasses a critical period of gliogenesis and cortical expansion. However, systematic single-cell analyses to resolve cellular diversity and gliogenic lineages of the third trimester are lacking. Here, we present a comprehensive single-nucleus RNA sequencing atlas of over 200,000 nuclei derived from the proliferative germinal matrix and laminating cortical plate of 15 prenatal, non-pathological postmortem samples from 17 to 41 gestational weeks, and 3 adult controls. This dataset captures prenatal gliogenesis with high temporal resolution and is provided as a resource for further interrogation. Our computational analysis resolves greater complexity of glial progenitors, including transient glial intermediate progenitor cell (gIPC) and nascent astrocyte populations in the third trimester of human gestation. We use lineage trajectory and RNA velocity inference to further characterize specific gIPC subpopulations preceding both oligodendrocyte (gIPC-O) and astrocyte (gIPC-A) lineage differentiation. We infer unique transcriptional drivers and biological pathways associated with each developmental state, validate gIPC-A and gIPC-O presence within the human germinal matrix and cortical plate in situ, and demonstrate gIPC states being recapitulated across adult and pediatric glioblastoma tumors.

Human neocortical development involves the concerted proliferation and lineage specification of progenitors in periventricular germinal matrix (GM) and the radial migration of their progeny to form the cortical plate (CP). Single-cell transcriptomics has contributed greatly to our understanding of neocortical development during the first and second trimesters of human gestation, facilitating the characterization of neurogenic progenitors between 12 and 28 gestational weeks (gw)[1–8]. Studies have identified important subtypes of radial glia (RG),

including early ventricular RG (vRG); truncated RG (tRG) that arise in the ventricular zone (VZ) after 18 gw and are distinguished from vRG by their truncated basal processes; and outer RG (oRG), which populate the outer subventricular zone (oSVZ)[2,4,9,10]. Similar analyses at mid-gestation of human development have also captured the presence of early neurogenic and gliogenic progenitors[1,3,4,11], including multipotent intermediate progenitor cells (MIPC) that are distinctive from tRG and give rise to both macroglia and interneurons[12]. Others have begun

[1]Department of Pathology, Molecular, and Cell-Based Medicine, Icahn School of Medicine at Mount Sinai, New York, NY 10029, USA. [2]Department of Neuroscience, Icahn School of Medicine at Mount Sinai, New York, NY 10029, USA. [3]Department of Neurology, Boston Children's Hospital, Boston, MA 02115, USA. [4]Department of Genetics and Genomic Sciences, Icahn School of Medicine at Mount Sinai, New York, NY 10029, USA. [5]Black Family Stem Cell Institute, Icahn School of Medicine at Mount Sinai, New York, NY 10029, USA. [6]These authors contributed equally: Susana I. Ramos, Zarmeen M. Mussa. ✉e-mail: Alexander.Tsankov@mssm.edu; Nadejda.Tsankova@mssm.edu

resolving oligodendrocyte progenitor cell (OPC) subtypes[13–16]. However, systematic transcriptomic analyses of human glial lineage specification during the third trimester, when glial populations expand, and astrogenesis begins, are lacking, in part due to limited availability of tissue samples from this period. As a result, our understanding of the molecular programs driving glial progenitor diversity and lineage choice during late corticogenesis is largely derived from animal studies[10,17–19]. Given the emerging role of glia in a myriad of neurodevelopmental, neuropsychiatric, neurodegenerative, and neoplastic disorders[20–26], a deeper understanding of their biology and molecular heterogeneity during normal human development is critical.

To gain further insight into prenatal gliogenesis, we sought to resolve progenitor cell types and lineage dynamics during the third trimester of human neocortical development through single-nucleus RNA sequencing (snRNA-seq)[27,28]. Our analyses uncover an epidermal growth factor receptor (EGFR)-positive glial progenitor population, gIPC, common to both oligodendrocyte and astrocyte lineages, as well as early OPC- and astrocyte-biased gIPC intermediate subpopulations emerging within the third trimester of neurodevelopment, with distinct transcriptomic profiles. We demonstrate the enrichment of specific prenatal glial cell-type signatures in several disease states, including recapitulation of multipotent and glial intermediate progenitor cell states across pediatric and adult glioblastoma tumors. Beyond this study, our resource dataset enables additional cell type-, region-, and gestational age-specific interrogation for novel markers and transcriptional signatures.

## Results

### Transcriptomic atlas of late prenatal neocortical development captures human gliogenesis with high temporal resolution

To resolve cell type diversity during human prenatal gliogenesis, we generated a snRNA-seq dataset from 15 fresh (unfixed) snap-frozen, anatomically intact, postmortem samples obtained from the second and third trimesters of gestation (17 to 41 gw) (Supplementary Data 1–2). We macrodissected and separately sequenced the germinal matrix, the primary site of gliogenesis, and the cortical plate, the migratory endpoint of differentiating neurons and glia (Fig. 1a). We also included an adult cohort to serve as a reference outgroup, dissecting both the subventricular zone (SVZ), a vestigium of the GM, with a fraction of the caudate nucleus, and the overlying posterior frontal lobe neocortex (CX) (Fig. 1a). Following quality control filtering and doublet removal (Methods, Supplementary Data 3–5), data from 178,580 prenatal and 21,934 adult nuclei were integrated[29] by age (prenatal or adult) and region (GM, CP, SVZ, or CX) and visualized using unsupervised uniform manifold approximation and projection (UMAP)-based embeddings[30] (Fig. 1b, c, Supplementary Fig. 1). For each integrated dataset, nuclei clustered according to distinct cell types and cell states rather than by age, sex, or postmortem interval (Supplementary Fig. 2), indicating that the clustering is driven primarily by biological variation rather than technical or batch effects.

Using canonical lineage and proliferation markers within the top differentially expressed genes (DEGs) per cluster[1–4,27,31–33], we annotated cell types and cell states across all four regions (Fig. 1b–e, g; Supplementary Fig. 1, Supplementary Data 6). We note greater cluster separation in the CP, SVZ, and CX, as compared to the GM, consistent with the more differentiated nature of their cell types (Fig. 1b, c, Supplementary Fig. 1a, c). Within the GM, nuclei cluster into 11 primary cell types (Fig. 1b), five of which are glial (Fig. 1d, Supplementary Data 6). We identify FAM107A+, FBXO32+ radial glia and astrocyte clusters (RG/AC; clusters 17, 27, 28), which are difficult to separate due to their similarity in marker expression; AQP4+, ALDH1L1+ astrocytes (cluster 14); and CSPG4+, PDGFRA+ OPCs (cluster 21). We also identify an EGFR+ glial intermediate progenitor cell (gIPC) population (cluster 9), which expresses both OPC and astrocyte progenitor markers, such as OLIG1[34]

and FGFR3[35]. Cluster 36 expresses ependymal cell marker, VWA3B, and choroid plexus marker, TTR. Clusters 29 and 34 comprise CX3CR1+ microglia and FLT1+ blood vessel cells (BVC), respectively. In the neuronal compartment, we identify EOMES+ neuronal intermediate progenitor cells (nIPC; clusters 13 and 32) and several clusters of maturing cortical projection neurons (CPN), including SATB2+, CUX2+ L2/3 CPN (clusters 2, 6, 7, 8, and 24), RORB+ L4/5a CPN (clusters 11, 16, 19, 20, and 31), and FOXP2+, FEZF2+ L5b/6 CPN (cluster 26) (Fig. 1d). We also resolve TLE4+, LMO3+ subplate neurons (SPN; clusters 22, 23, and 35), maturing TAC1+, PENK+ medium spiny neurons (MSN; clusters 18 and 30), GAD1+ and DLX6-AS1+ interneurons (IN; clusters 1, 3, 4, 5, 10, 25, and 33), as well as an undefined population (UD) marked by high expression of tubulins and ribosomal genes, rather than by a specific cell lineage (clusters 0, 15, 37) (Fig. 1d, Supplementary Data 6). Cluster 12 consists of MKI67+, TOP2A+ cycling or transit-amplifying cells (TAC) (Fig. 1d, g), thought to represent a cellular state of actively dividing cells, rather than a specific cell type.

Similar cell types and states are identified in the CP, with greater granularity for neuronal types (Fig. 1c, e, g). Most EOMES+ nIPCs in the CP (cluster 5) likely result from the inclusion of the intermediate zone and part of the oSVZ in younger samples, and cluster 9 contains markers of both oRGs and astrocytes (Fig. 1e). The CP also includes RELN+ Cajal-Retzius cells (CRN; clusters 28 and 31). As expected, maturing MSN, ependymal cells, and choroid plexus cells are absent in the CP.

Integrated analysis of the adult SVZ identifies nine cell types (Supplementary Fig. 1b). MSN of the striatum can be separated into TAC1+, CHRM4+ D1-type MSN (clusters 8, 13, and 17) and PENK+, GPR6+ D2-type MSN (clusters 3, 6 and 15). Interneurons divide into five clusters. Cluster 11 is defined by PVALB expression; clusters 14 and 18 by HTR3A, CCK, CALB2, and VIP; cluster 19 by SST, NOS1, and NYP; and cluster 25 by CALB2 expression. SATB2+ excitatory neurons (EN; cluster 21) are also identified. GFAP+, SLC1A3+ (protoplasmic) astrocytes (cluster 7) are distinct from GFAP+, CD44+ (fibrous) astrocytes (cluster 16) as well as from VWA3B+ ependymal cells (cluster 24). We also identify PDGFRA+, CSPG4+ OPCs (cluster 5), MBP+, PLP1+ oligodendrocytes (OL; clusters 0, 1, 2, 10, and 23), CX3CR1+ microglia (MG; cluster 4), FLT1+ BVCs (cluster 22), and CD2+ T-cell lymphocytes (cluster 20). Clusters 9 and 12 could not be classified based on canonical lineage markers, and are marked by the upregulation of long noncoding RNA genes (Supplementary Data 6). The adult CX includes the aforementioned cell types with the exception of striatal MSNs, and further resolves a BCAS1+ population of pre-myelinating / early myelinating oligodendrocytes[36] (preOL; cluster 27) (Supplementary Fig. 1c, d).

We next leveraged the high number of nuclei and developmental time points sampled in our dataset to quantify cell types across gestational age. Focusing on the germinal matrix where gliogenesis begins, samples were grouped into four stages spanning the latter half of second trimester and third trimester: 17–20 gw (Stage 1), 20–24 gw (Stage 2), 24–28 gw (Stage 3), and 31–41 gw (Stage 4). The stages differed statistically by gestational age but not by sex or tissue collection postmortem interval (Supplementary Data 1–2). By plotting mean fraction of cell type by stage, we observe two trends from second to third trimester: a decrease in the fraction of nIPCs and excitatory neurons (EN: CPN and SPN) and an increase in the fraction of glia (RG/AC, gIPC, astrocyte, OPC, and microglia) (Fig. 1f). To study these trends further, we correlated the fraction of cell types to sample age in gestational weeks (Supplementary Fig. 3). This reveals a significant increase with gestational age in the fraction of gIPCs ($p = 0.003$; two-tailed Spearman Rank-Order Correlation test), astrocytes ($p = 0.015$), OPCs ($p = 0.0008$), and microglia ($p = 0.002$) (Supplementary Fig. 3a). The fraction of MSNs also increase significantly ($p = 0.002$). Conversely, there is a significant decrease in the fraction of nIPCs ($p = 0.034$), late-born L2/3 CPNs ($p = 0.009$), and TAC ($p = 0.002$)

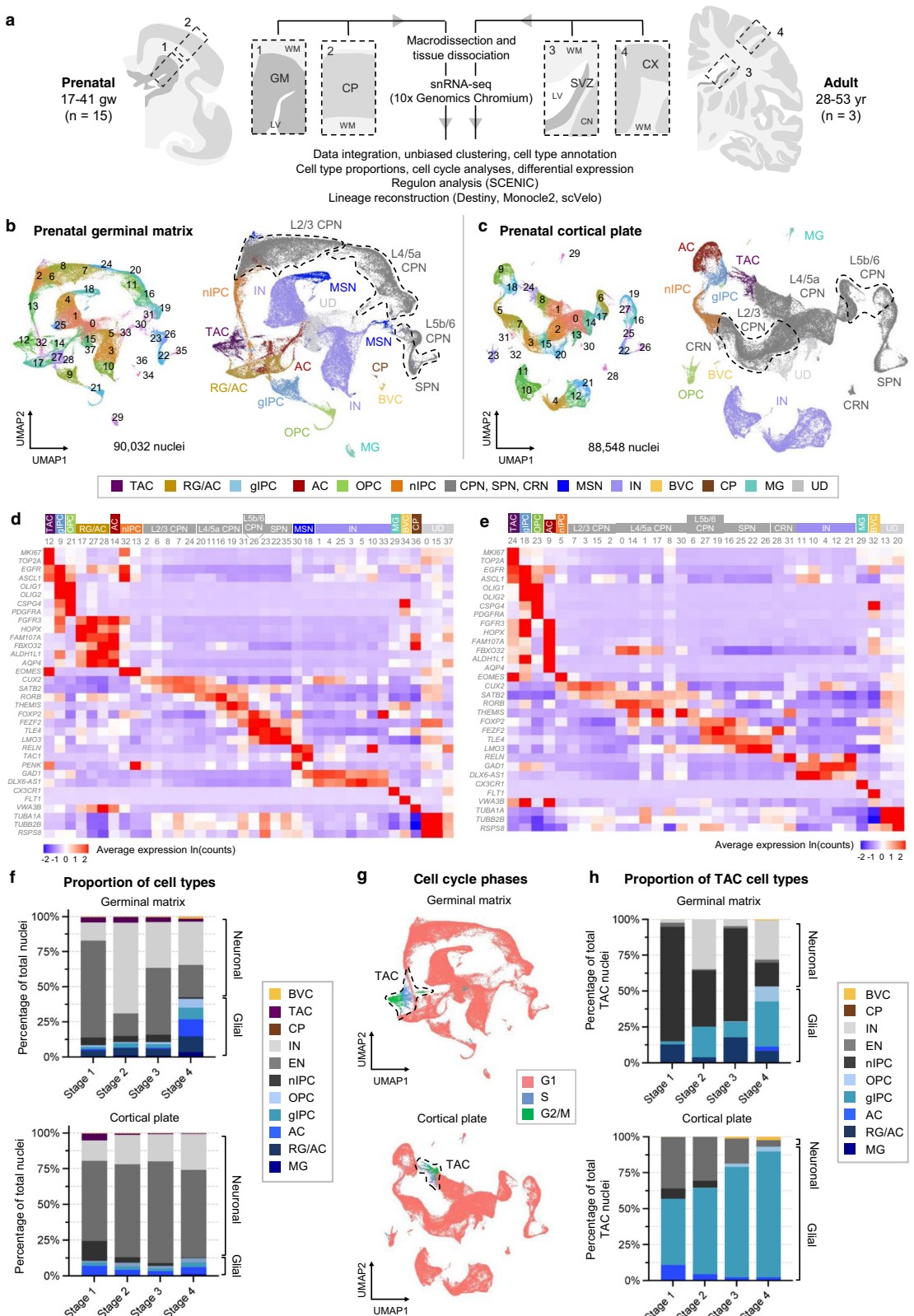

**f** Proportion of cell types **g** Cell cycle phases **h** Proportion of TAC cell types

(Supplementary Fig. 3a). Our GM dataset, therefore, captures the start of the gliogenic period. In the CP, the fraction of TACs, nIPCs, and CRNs exhibit a significant decrease over time ($p = 0.002$, ~0, 0.0001, respectively), while L4/5a CPNs show a significant increase ($p = 0.002$) (Supplementary Fig. 3b). Interneurons also increase from second to third trimester ($p = 0.001$), consistent with their migration into the neocortex. While the fraction of CP glia exhibit a positive trend, only

OPC numbers show a significant positive correlation with gestational age ($p = 0.026$).

To study cell cycle dynamics over the late gestational period, we characterized the cell cycle phase of each cell type[37] using a curated list of cell cycle marker genes[38]. Predictably, this reveals enrichment of G2/M- and S-phase proliferative signatures in TACs (Fig. 1g), but also within a fraction of proliferative glia (2.29% of RG/AC in the GM; 5.34%

**Fig. 1 | Single nucleus transcriptomic atlas of prenatal and adult human brain regions. a** Schematic of macrodissection and sequencing approach to study germinal matrix (GM, $n = 15$) and cortical plate (CP, $n = 15$) development in the second and third trimesters (17–41 gw) with corresponding adult brain regions, SVZ ($n = 3$) and CX ($n = 3$). **b, c** UMAP plots for all prenatal samples integrated by anatomical region: GM (**b**) and CP (**c**). Clusters (left) are colored by cell type annotation (right). **d, e** Heatmaps of log-normalized average gene expression per cluster, showing selected canonical marker genes (rows) used to assign cell identity to each cluster (columns) in (**b**) and (**c**), respectively. Clusters are grouped and colored by cell identity. **f** Stacked bar plots of stage-normalized cell type proportions in the GM (top) and CP (bottom), showing changes in the factions of glia and neurons over four defined stages of prenatal development: stage 1 (17–20 gw), stage 2 (20–24 gw), stage 3 (24–28 gw), stage 4 (31–41gw). Normalization is based on the total number of nuclei assigned to a cell type. CPN, SPN, and CRN included as part of EN, MSN included as part of IN, UD not included. **g** UMAP display of cell cycle phase annotations within the integrated GM (top) and CP (bottom), highlighting

actively cycling cells as TAC. **h** Stacked bar plot of TAC state cell type proportions in GM (top) and CP (bottom). Cell type identity is based on prediction scores from cell type signatures in (**b**) and (**c**). Source data for (**f, h**) are provided as a Source Data file. See also Supplementary Figs. 1–3. TAC transit-amplifying cell/cycling progenitor, RG radial glia, oRG outer radial glia, tRG truncated radial glia, EPD ependymal cell, AC astrocyte, AC-f fibrous astrocyte, AC-p protoplasmic astrocyte, gIPC glial intermediate progenitor cell, OPC oligodendrocyte progenitor cell, preOL premyelinating/early myelinating *BCAS1*+ oligodendrocyte, OL oligodendrocyte, nIPC neuronal intermediate progenitor cell, mIPC multipotent intermediate progenitor cell, UD undefined, MSN medium spiny neuron, EN excitatory neuron, CPN cortical projection neuron, SPN subplate neuron, CRN Cajal Retzius cell, IN interneuron, MG microglia, BVC blood vessel cell, L2/3, L4, L5/6 neocortical layers 2/3, 4, 5/6, GM or G prenatal germinal matrix, CP prenatal cortical plate or choroid plexus, SVZ adult subventricular zone, CN caudate nucleus, LV lateral ventricles, WM white matter, CX adult neocortex, gw gestational weeks, yr years.

of gIPCs in the GM and 5.84% in the CP; 6.15% of OPC in GM and 5.90% in the CP; 2.03% of microglia in the GM and 4.41% in the CP) (Supplementary Fig. 3c). nIPCs only show high G2/M- and S-phase scores in the GM (4.51% of nIPCs in the GM versus 0.65% in the CP), demonstrating how these progenitors exit the cell cycle as they migrate through the germinal matrix. To further characterize TACs (GM cluster 12 in Fig. 1b, d and CP cluster 24 in Fig. 1c, e), we regressed out G2/M- and S-phase signatures and subclassified TACs using our annotated cell types as a reference (Fig. 1h). In the GM, this analysis highlights nIPC as the main proliferative cell type in the second trimester (80.00%, 39.31%, and 64.80% from Stages 1–3, respectively). By the third trimester, gIPCs become the most proliferative cell type (Stage 4 = 31.59%). In the CP, gIPCs make up the majority of TACs (Fig. 1h). By stage 4 (31–41gw), they represent >85% of cycling cells.

## Resolving the diversity of progenitor and glial subtypes in the germinal matrix

As our snRNA-seq dataset uniquely captures the expansion of gliogenesis during late prenatal development, we focused subsequent analyses on further resolving glial diversity and lineage relationships. To gain better cell type separation, we subclustered glial and progenitor populations (astrocyte, RG/AC, gIPC, OPC, and nIPC) (Fig. 2). TACs were also included but their cell cycle signature was regressed out in order to redistribute them into their respective cell types (Fig. 1h). Using canonical and de novo differential gene expression analysis (DGEA) markers, we further resolve RG into distinct *HOPX*+ oRG (subcluster 5) and *PALLD*+ tRG (subcluster 32). We annotate *EGFR +, OLIG1+, FGFR3*+ gIPC populations (subclusters 3, 11, 16, 26, 30), *AQP4*+ astrocytes (subclusters 4, 6, 14, and 24), *VWA3B*+ ependymal cells (subcluster 23), *TTR*+ choroid plexus cells (subcluster 33), nIPCs (subclusters 1, 12, 15, 17, 18, 20, 21, and 25), *CUX2*+ CPNs (subcluster 31), and *GAD1*+ interneurons (subcluster 22) (Fig. 2a, b), the latter two likely emerging from the subclustering of nIPCs or TACs. A subset of RG and AC are still annotated as RG/AC (subclusters 0, 2, 9, and 27) due to their abundance in both earlier stages 1–2 (expressing RG markers) and in later stages 3–4 (expressing AC markers) (Supplementary Fig. 4a). Notably, the subclustered analysis also resolves a recently characterized multipotent intermediate progenitor cell (mIPC) population[12,19] (subclusters 7, 10, and 13), expressing proliferation markers *TOP2A* and *MKI67* and low levels of *EGFR* and *ASCL1*. mIPCs are most abundant in the earlier stages of our analysis and cluster separately from later appearing gIPCs (Supplementary Fig. 4a).

Given the overlap between some mIPC and gIPC markers[12,19] (Fig. 2b), we performed DGEA comparing the two populations to better define their differences (Fig. 2c, Supplementary Data 7). This analysis reveals that mIPCs are enriched for both neuronal and radial glial markers, including *RBFOX1, ADGRV1,* and *NRG1,* as well as for cell cycle genes. In contrast, gIPCs are enriched for glial markers, including

*OLIG1/2* and *SLC1A2,* and markers associated with migration, including *ERBB4* and *PCDH15*[15,39] (Fig. 2c, Supplementary Data 7). Notably, *EGFR* is a differential marker for gIPCs. Gene set enrichment analysis corroborates these findings. mIPCs show enrichment for gene ontology (GO) terms such as "G2/M cell transition" and "positive regulation of neural precursor proliferation", while gIPCs are enriched for "cell-cell adhesion," "chemotaxis," and "positive regulation of cell migration" (Fig. 2d, Supplementary Data 8–9). Notably, maturation analysis using signatures from terminally differentiated cell types (top 50 DEGs for astrocytes, oligodendrocytes, excitatory neurons, and interneurons) reveals mIPCs to be less differentiated than gIPCs along the glial lineage ($p < 0.05$, Wilcoxon Rank Sum test) (Supplementary Fig. 4b–d). mIPCs are, however, more differentiated than gIPCs along the neuronal lineage, consistent with their proposed role as neuronal progenitors[12,19].

We then focused our analysis on resolving gIPC heterogeneity. Looking at the five gIPC subclusters (Fig. 2a), gIPC26 clusters closest to OPCs while gIPC30 is adjacent to astrocytes. To examine potential lineage bias, we calculated astrocyte and OPC scores for each gIPC and mIPC subpopulation, using the top 50 DEGs for astrocytes and OPCs. We find that gIPC30 is most significantly enriched for the GM astrocyte score ($p = 1.05E-90$, 2.06E-26, 6.79E-73, 3.43E-87, 8.02E-69, 7.97E-59, and 2.12E-103 for comparisons with mIPC7, mIPC10, mIPC13, gIPC3, gIPC11, gIPC16, and gIPC-O26, respectively; Pairwise Wilcoxon Rank Sum with Holm correction) (Fig. 2e) while gIPC26 is most significantly enriched for the computed GM OPC score ($p = 6.55E-144$, 1.44E-101, 5.27E-102, 3.09E-104, 4.64E-102, 1.21E-37, and 1.21E-71 for comparisons with mIPC7, mIPC10, mIPC13, gIPC3, gIPC11, gIPC16, gIPC-A30, respectively), (Fig. 2f). As a result, we annotated gIPC26 as an oligodendrogenesis-biased gIPC (gIPC-O) and gIPC30 as an astrogenesis-biased gIPC (gIPC-A). Remaining gIPC subclusters 3, 11, and 16 also exhibit higher OPC and astrocyte median scores compared to mIPC subclusters 7, 10, and 13, consistent with their respective glial versus multipotent progenitor designations (Fig. 2e–f, Supplementary Fig. 4c). Notably, mIPC13 exhibits the highest median score for the interneuron signature ($p = 4.14E-57$, 1.68E-05, 6.52E-62, 1.26E-51, 8.38E-22, 2.35E-08, and 3.40E-13 for comparisons with mIPC7, mIPC10, gIPC3, gIPC11, gIPC16, gIPC-O26, and gIPC-A30, respectively) (Supplementary Fig. 4d), consistent with its proximity to interneuron subcluster 22 and predicted neurogenic potential (Fig. 2a). We also looked at lineage marker expression across all four developmental stages. Among gIPC subtypes, gIPC-O and gIPC express the highest levels of *EGFR*, and gIPC-O expresses the highest level of *PCDH15* and *OLIG2* in a pattern that parallels that of OPCs (Supplementary Fig. 4e). gIPC-A express the highest levels of astrocyte lineage markers *SOX9* and *AQP4*, in a similar pattern to astrocytes, and the lowest levels of *EGFR* and *OLIG2* markers (Supplementary Fig. 4e). By quantitating their proportions over time, we observe significant increases in the fractions of gIPC, gIPC-A and

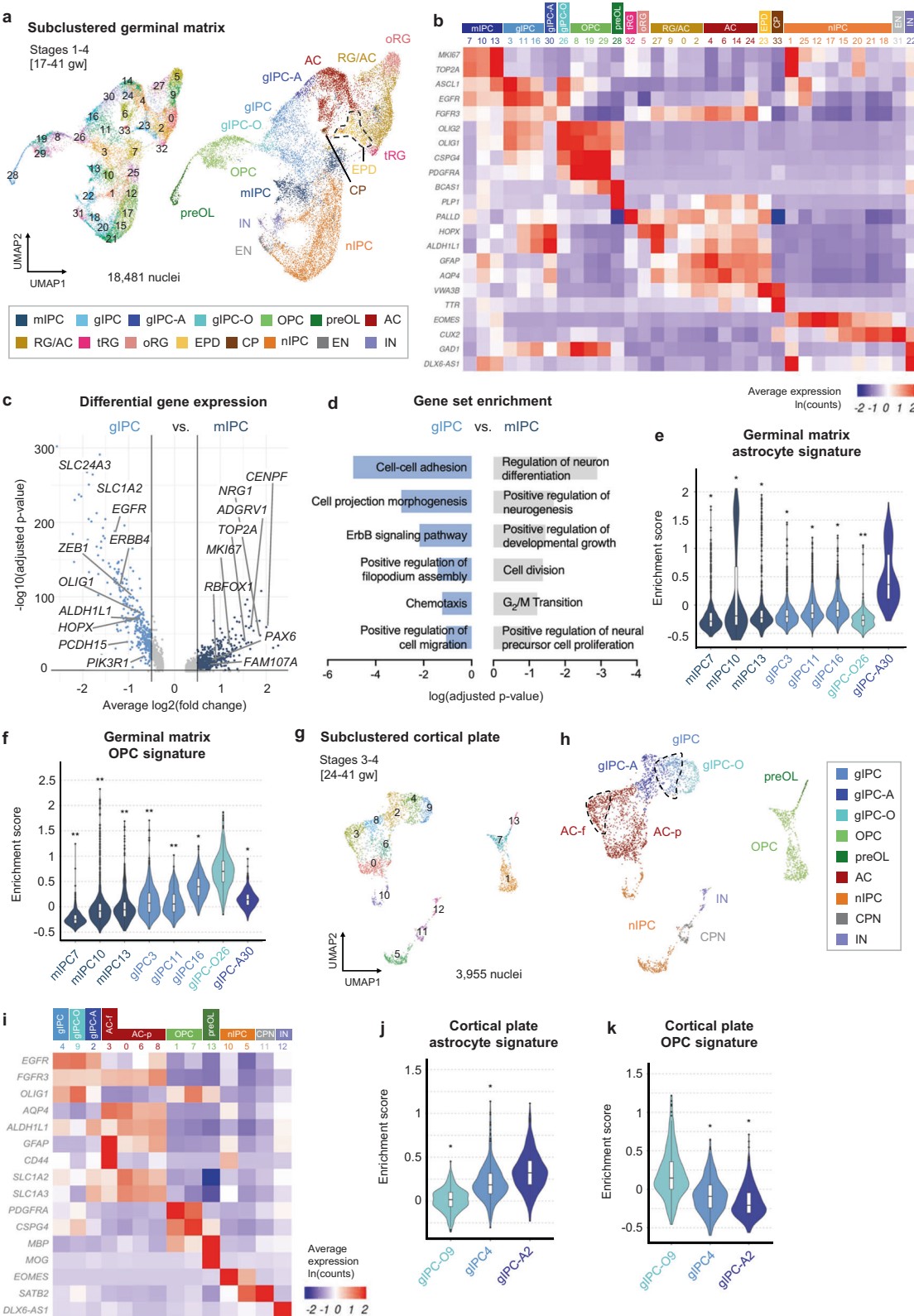

gIPC-O subpopulations ($p = 0.008$, 0.023, and 0.002, respectively; two-tailed Spearman Rank-Order Correlation test) (Supplementary Fig. 5a). By contrast, mIPCs and oRG show a significant decrease ($p = 0.002$ and 0.001, respectively).

To understand how our progenitor annotations relate to those in recent literature, we performed correspondence analysis between our data and recently published mid-gestation human and mouse neurodevelopmental datasets[12,16,19]. We used the published annotations to train a classifier and transferred the reference annotation labels onto our data (Supplementary Fig. 6a–e). The closest overall correspondence is observed between our early neurogenic period (stages 1–2) dataset and the recent Yang et al. (2022) human study[12] of a similar gestational period (18–23 gw), which first describes human multipotent IPCs (bMIPCs, Supplementary Fig. 6b). Yang et al.

**Fig. 2 | Resolving glial diversity in the prenatal germinal matrix and cortical plate. a** UMAP plots of glia, nIPC, and cell cycle regressed TAC isolated from Fig. 1b for subclustered analysis. Clusters (left) are colored by cell subtype annotation (right). **b** Heatmap of log-normalized average gene expression, showing canonical marker genes used to assign cell identity to each cluster in (a). **c** Differential gene expression analysis between GM gIPC and mIPC populations. Significance determined by two-sided Wilcoxon Rank Sum test (p-adj. >0.05; average log2(fold change) >0.5 or < −0.5). **d** Gene set enrichment analysis, showing top curated terms enriched in gIPC and mIPC. Significance determined by hypergeometric test with Benjamini-Hochberg correction (p-adj. >0.05). All terms are shown in Supplementary Data 8–9. **e, f** Violin plots showing germinal matrix astrocyte and OPC scores in mIPC and gIPC subclusters. Enrichment scores are computed using the top 50 DEGs for astrocytes (subclusters 4, 6, 14, and 24) and OPCs (subclusters 8, 19, and 29). **g, h** UMAP plots of cortical plate glia, n-IPCs, and cell cycle regressed TACs, isolated from 1c for subclustered analysis. Stages 1 and 2 excluded to minimize cross-contamination from the GM. Plots are colored by cluster number (g) or by cell subtype annotation (h). **i** Heatmap of average log-normalized gene expression,

showing canonical marker genes used to assign cell identity to each cluster in (g–h). **j, k** Violin plots showing cortical plate astrocyte and OPC scores in gIPC subclusters. Enrichment scores are computed using the top 50 DEGs for astrocytes (subclusters 0, 3, 6, and 8) and OPCs (subclusters 1 and 7). Statistically significant differences between subclusters in (e, f, j, and k) are determined using pairwise two-sided Wilcoxon Rank Sum tests with Holm correction (*$p < 1e^{-5}$, ** $p < 1e^{-100}$, ***$p < 1e^{-200}$, and ****$p < 1e^{-300}$), showing the significance between the highest scoring population against all others. Box plots in (e, f, j, k) show the median, bounds of box represents the first and third quartile, whiskers extend to the minimum and maximum values within 1.5 times of the interquartile range. Source data for (c–f, j, k) are provided as a Source Data file. See also Supplementary Figs. 4 and 5. RG radial glia, oRG outer radial glia, tRG truncated radial glia, EPD ependymal cell, AC astrocyte, AC-f fibrous astrocyte, AC-p protoplasmic astrocyte, gIPC glial intermediate progenitor cell, OPC oligodendrocyte progenitor cell, preOL premyelinating/early myelinating *BCAS1+* oligodendrocyte, nIPC neuronal intermediate progenitor cell, mIPC multipotent intermediate progenitor cell, EN excitatory neuron, CPN cortical projection neuron, IN interneuron, CP choroid plexus, gw gestational weeks.

EGFR+/ASCL1+/OLIG1+/OLIG2+ bMIPC match to our mIPC, gIPC, and gIPC-O; their SPARCL1+/OLIG1+/OLIG2+ APC match our gIPC-A and astrocytes; and their OPC match our gIPC-O, OPC, and premyelinating oligodendrocytes (Supplementary Fig. 6a–c). In the gliogenic period (stages 3–4), their bMIPCs match most closely with our gIPCs (Supplementary Fig. 6c). Compared to the Fu et al. dataset[16], we find concordance between their cycling cells and our highly proliferative mIPC population (Supplementary Fig. 6d). Moreover, we find close matches between their SPARCL1+/HOPX+ oAPC, a hybrid progenitor that co-expresses oligodendrocyte and astrocyte markers, and our gIPC-A, as well as their EGFR+/OLIG2+ priOPC with our gIPC and gIPC-O sub-populations (Supplementary Fig. 6d). The mouse bMIPC signature from the Li et al. dataset[19] does not match closely with our multipotent and gliogenic progenitors, showing potential species-specific differences (Supplementary Fig. 6e). Overall, this analysis underscores the unique glial resolution of our late gestation prenatal dataset.

## Resolving differentiating glial cell types in the cortical plate

To expand our understanding of glial progenitor diversity beyond the germinal matrix, we subclustered cell cycle-regressed macroglia and progenitors from the cortical plate. We focused our analyses on late prenatal stages 3 and 4 to avoid potential cross-contamination from the GM and resolved numerous gIPC, astrocyte, and OPC sub-clusters using a similar canonical marker annotation approach used for the GM (Fig. 2g–i). Of note, CP astrocytes show better separation into protoplasmic and fibrous subtypes based on adult marker expression (Fig. 2i). Unlike the GM, we do not resolve a distinct mIPC cluster in the CP, suggesting regional specificity. In contrast, gIPCs are prevalent in the cortical plate (subcluster 4 in Fig. 2g–h). Similarly to the GM, we also define gIPC-A and gIPC-O subpopulations in the cortical plate (subclusters 2 and 9, respectively) based on their significant enrichment for astrocyte ($p = 5.45E-65$ and $4.33E-16$ for comparisons with gIPC-O9 and gIPC4, respectively; Pairwise Wilcoxon Rank Sum with Holm correction) and OPC scores ($p = 1.56E-25$ and $5.62E-47$ for comparisons with gIPC4 and gIPC-A2), respectively (Fig. 2j, k). Their proportions, however, do not change over time (Supplementary Fig. 5b).

By performing immunofluorescence analysis, we verified the presence of gIPCs in both the germinal matrix and cortical plate at 20 and 30 weeks of gestation (Fig. 3a–f). Consistent with previous reports[12,40], we find VZ enrichment of EGFR+ cells at 20 gw (Fig. 3a, left). Here, they extend radial processes reminiscent of tRG. By 30 gw, however, few EGFR+ cells are found in the VZ/iSVZ (Fig. 3a, right). We find that they are most prevalent in the oSVZ where they exhibit uni- or bipolar morphology and extend tangential processes, a phenotype of migratory cells (Fig. 3b–d). In the CP, EGFR+ cells are prevalent in neocortical layers 5/6 (Fig. 3e–f). In both regions, we find

colocalization of EGFR with the oligodendrocyte lineage marker, OLIG2, and/or the astrocyte marker, SOX9, at 30 gw (Fig. 3c, d, f). This aligns with our transcriptomic findings of *EGFR+/OLIG2+/SOX9−* gIPC-O and *EGFR+/SOX9+/OLIG2−* gIPC-A (Fig. 2b, i; Supplementary Fig. 4e). We do not observe differences in the distribution of these phenotypes in either region.

Finally, we sought to compare our prenatal cell types to their adult counterparts. We trained a classifier on the higher resolution prenatal cell type identities annotated in our subclustered analysis (Fig. 2) and assessed the correspondence to cell identities from the adult SVZ or CX (Supplementary Fig. 1). We find that prenatal oligodendroglial, ependyma, vascular, microglia, and most neuronal populations all have strong correspondence scores with their adult counterparts (Fig. 3h, i). Interestingly, prenatal astrocytes from the GM match more strongly with adult fibrous astrocytes, AC-f, than with adult proto-plasmic astrocytes, AC-p (Fig. 3h). Performing the same analysis using CP annotations against known adult CX cell types, we find overall similar results with the exception of strong correspondence of pre-natal CP astrocytes to both adult AC-f and AC-p (Fig. 3i). Notably, we do not find any correspondence between prenatal gIPC, gIPC-A, gIPC-O, mIPC, tRG, or oRG and adult cell types, arguing that, similarly to radial glia and nIPCs[2,4,9,10], mIPCs and gIPCs also represent transient neuro-developmental populations.

## Transcriptomic analysis of biological drivers in gIPC-A and gIPC-O cell identity

To understand further the biology of gIPC subtypes, we compared the transcriptional differences between gIPC-A and gIPC-O populations in the GM (Fig. 4a). DGEA shows differential expression of OPC markers (*LHFPL3* and *PCDH15*) in gIPC-O and astrocyte markers (*AQP4*, *SOX9*, and *GFAP*) in gIPC-A (Fig. 4b, Supplementary Data 7). Notably, gIPC markers *OLIG1/2* are significantly enriched in gIPC-O, indicating divergence of the gIPC-A lineage. Moreover, this analysis identifies transcription factor (TF) genes *ZEB1* and *FOXO1* as potential markers for each population. Functional gene set enrichment analysis on the top differentially expressed gIPC-A vs. gIPC-O genes shows significant enrichment of some pathways in common to both gIPC subpopula-tions, as well as significant enrichment of unique pathways for each (Fig. 4c, Supplementary Data 10–11). These include WNT signaling pathway unique to gIPC-A, and NOTCH signaling pathway unique to gIPC-O (Fig. 4c, Supplementary Data 10–11). Regulon analysis using SCENIC to infer gene modules and TF drivers in gIPC subsets (Fig. 4d, e) further supports the above findings. ZEB1 is highlighted as an oligodendroglial lineage-specific TF regulator (Supplementary Fig. 4f, Supplementary Fig. 6f), and its associated regulon displays high activity in gIPC, gIPC-O, and OPC populations (Fig. 4e). The inferred regulon of *FOXO1*, by contrast, has high activity in gIPC-A and astrocyte

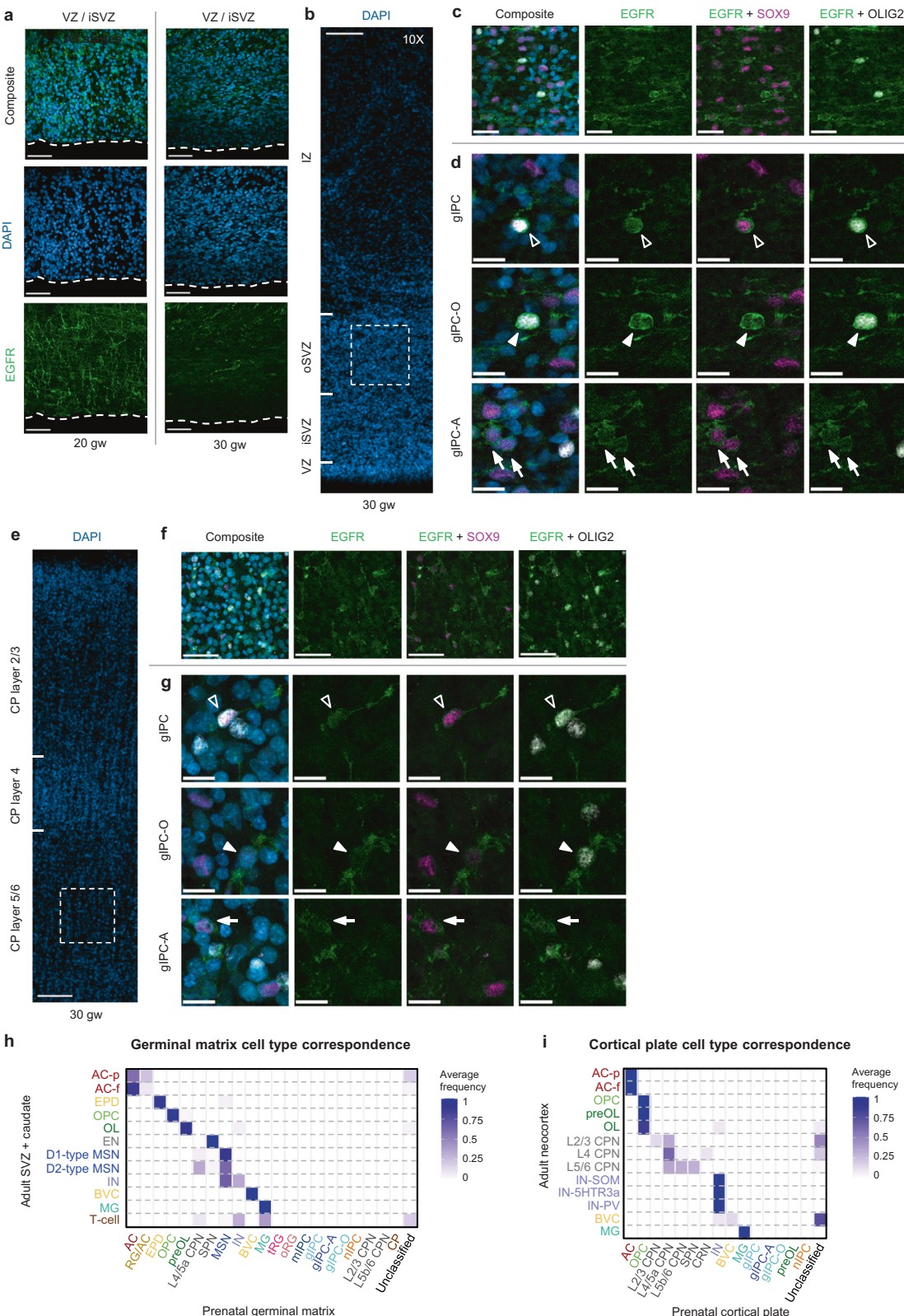

populations (Fig. 4e). At the intersection of *ZEB1* regulon genes (81 genes; Fig. 4f) and gIPC-O DEGs (191 genes; p-adj. <0.05), we find NOTCH pathway-signaling genes (*DLL1*, *DLL3*, and *NCOR2*), among others (Fig. 4g). In genes shared between *FOXO1* regulon genes (15 genes; Fig. 4h) and gIPC-A DEGs (1034 genes, p-adj. <0.05) we identify WNT pathway regulatory targets (Fig. 4i). Through immunohisto-chemical analysis, we confirm the presence of EGFR+/ZEB1+ cells in the

human oSVZ and IZ at 30 gw, which represent the majority of EGFR+ glia in these regions (Fig. 4j).

## Computational reconstruction of gIPC lineages

Having characterized gIPCs, gIPC-A, and gIPC-O as key gliogenic populations, we next sought to infer developmental hierarchies between progenitor and glial cell types using several computational

**Fig. 3 | Histological validation of glial progenitor subtypes.**
**a** Immunofluorescence analysis for EGFR (green) at the ventricular zone in second (20 gw) and third (30 gw) trimester GM tissue samples. Scale bar = 50 μm.
**b**–**g** Immunofluorescence analysis for EGFR (green), SOX9 (magenta), and OLIG2 (grey) at the oSVZ of the human GM (**b**–**d**) and CP (**e**–**g**) in third (30 gw) trimester tissue sample. Low magnification image of DAPI staining, showing the subdivisions of the GM (**b**) and CP (**e**). Scale bar = 100 μm. High magnification z-stack projections of the oSVZ (**c**) (boxed area in **b**) and layer 5/6 (**f**) (boxed area in **e**), revealing gIPC morphology (orientation and polarity). Scale bar = 50 μm. Single z-plane images (**d**, **g**) showing genetic marker colocalization in gIPC (EGFR+/OLIG2+/SOX9+; open arrowhead), gIPC-O (EGFR+/OLIG2+/SOX9–; closed arrowhead), and gIPC-A sub-types (EGFR+/OLIG2–/SOX9+; closed arrow). Scale bar = 15 μm. Immuno-fluorescence experiments in (**a**–**g**) were repeated at least three independent times, with similar results. **h**–**i** Confusion matrices showing the frequency of identity prediction matches between prenatal (x-axis) and adult (y-axis) SVZ+ caudate (**h**) or neocortex (**i**) cell identities. Nuclei with maximum prediction scores of <0.5 are marked as unclassified. Source data for (**h**–**i**) are provided as a Source Data file. RG radial glia, oRG outer radial glia, tRG truncated radial glia, EPD ependymal cell, AC astrocyte, AC-f fibrous astrocyte, AC-p protoplasmic astrocyte, gIPC glial inter-mediate progenitor cell, OPC oligodendrocyte progenitor cell, preOL pre-myelinating/early myelinating *BCAS1*+ oligodendrocyte, OL oligodendrocyte, nIPC neuronal intermediate progenitor cell, mIPC multipotent intermediate progenitor cell, UD undefined, MSN medium spiny neuron, EN excitatory neuron, CPN cortical projection neuron, SPN subplate neuron, CRN Cajal Retzius cell, IN interneuron, MG microglia, BVC blood vessel cell, L2/3, L4, L5/6 neocortical layers 2/3, 4, 5/6, GM prenatal germinal matrix, CP prenatal cortical plate or choroid plexus, i/oSVZ inner/outer subventricular zone, VZ ventricular zone, IZ intermediate zone, gw gestational weeks.

tools designed for trajectory analysis. First, we used Monocle2[6,41,42], an algorithm that orders cells along pseudotime corresponding to infer-red biological processes and lineage trajectories. In the GM, we find that our prenatal glia and progenitors form a lineage trajectory with nIPC and RG populations present in early stages and gIPCs, OPC, and astrocytes appearing in later prenatal stages (Fig. 5a–e, Supplementary Fig. 7a, b). Consistent with the expected temporal progression of the lineages seen, tRG, oRG, and nIPC populations display the lowest pseudotime and astrocytes display the highest (Fig. 5b). As pseudo-time progresses, RG feed into gIPCs, which, in turn, form distinct branches of gIPC-O to OPCs and gIPC-A to astrocytes (Fig. 5c–e). These branches exhibit high expression of *ZEB1* and *FOXO1*, respectively (Fig. 5d, e, right). We corroborate these observations using diffusion maps, a dimentionality reduction method used for representing cell state transitions in developmental lineage trajectories[43] (Supplemen-tary Fig. 7c). In late prenatal stage 3–4 (24–41 gw) diffusion maps reveal three apexes with mIPCs and gIPCs in the center (Supplementary Fig. 7c). mIPC positioned next to tRG appear to feed into the nIPC branch as well as into the gIPC population.

We also leveraged transcript splicing information to infer the directionality of cell type differentiation using scVelo[44–46]. Since dif-ferentiation dynamics are time point and region-specific, we mapped velocity vectors onto UMAP embeddings per stage (Fig. 5f–i, Supple-mentary Fig. 8). In the GM, we find a consistent pattern of differ-entiation across stages, in which vectors typically originate from cell cycle-regressed mIPCs (Fig. 5f–i). Analysis of earlier (predominantly neurogenic) stages 1 and 2 (17–24gw) reveal mIPCs (neighboring tRG) as a central population with velocity vectors pointing towards both neurogenic and gliogenic clusters (Fig. 5f–g). Within GM stage 1 (17–20 gw), we find three inferred lineage directionalities: mIPC to nIPC to CPN; mIPC to oRG; and gIPC to OPC (Fig. 5f). Even at this early time point, we observe the simultaneous induction of *OLIG1* and *FGFR3* in gIPCs, indicating bi-directionality (Supplementary Fig. 8a). Within GM stage 2 (20–24 gw), we also note velocity vectors directed from mIPC to gIPC, indicating potential origin for gIPCs. While nIPC to CPN and gIPC to OPC lineages remain prevalent at stage 2, we begin to resolve distinctive gIPC to astrocyte lineage (Fig. 5g). *FGFR3* expression is induced in a subset of gIPCs, which downregulate *OLIG1* expression in an apparent switch from oligodendrogenesis to astrogenesis (Sup-plementary Fig. 8b).

For scVelo Stage 3–4 analysis, we focused on our subclustered progenitor objects, which capture gliogenesis and define gIPC sub-types with superior resolution. Within subclustered stage 3, we observe clearly defined velocity vectors from gIPC to gIPC-A to astrocytes (Fig. 5h), as well as from gIPC to gIPC-O to OPC in stages 3 and 4 (Fig. 5h, i). *FGFR3* induction is seen in gIPC-A whose vectors point towards astrocytes, and *OLIG1* induction is seen in gIPC and gIPC-O (Supplementary Fig. 8c, d). Velocity of *ZEB1* is highest within gIPC-O clusters while velocity of *FOXO1* is highest in gIPC-A and astrocyte clusters (Fig. 5j, k), further implicating these developmental TF genes

in the biology of their respective gIPC subpopulations. Overall, this analysis infers neuronal/glial bidirectionality of mIPCs during earlier gestation, with tRG/mIPCs giving rise to a glial-only gIPC intermediate that directs lineage differentiation for both OPCs (via gIPC-O inter-mediate) and for astrocytes (via gIPC-A intermediate).

Stage-specific Monocle2 (Fig. 6a–f) and diffusion mapping ana-lyses (Supplementary Fig. 7d) in the cortical plate further support the role of gIPC as a bidirectional glial progenitor population. In stages 3–4 (Fig. 6a, d), gIPCs exhibit the lowest pseudotime and branch into gIPC-A and gIPC-O lineages (Fig. 6b, e). We find velocity vectors pointing from gIPC-O towards OPCs and from gIPC-A towards astrocytes in stage 3–4 samples (Fig. 6g, h). In both stages, gIPCs cluster with and point primarily towards astrocytes. As observed in the GM, gIPCs in the CP are consistently marked by *EGFR* expression (Supplementary Fig. 8e, f). Preferential expression of *FOXO1* in gIPC-A and astrocytes, and of *ZEB1* in gIPC, gIPC-O and OPC, is seen again (Fig. 6c, f).

## Association of prenatal signatures with human disease

Finally, we studied the relevance of the herein-resolved middle and late human prenatal developmental signatures in the context of disease. We calculated enrichment scores for several developmental disorders (including malformation of cortical development (MCD) I, II, III[47,48]; epilepsy; intellectual disability (ID); autism spectrum disorder (ASD)[3]; and glioma[49] (Fig. 7a, b, Supplementary Fig. 9). In addition to the enrichment of specific neuronal subpopulations in epilepsy and autism spectrum disorders, as previously seen in mid-gestation[3], this analysis discovers enrichment of gIPC, prenatal astrocyte, OPC, and radial glia subpopulations in association with ASD, ID, MCD, and gliomas (Fig. 7a, Supplementary Fig. 9a–d). Notably, gIPCs are most strongly enriched for genes associated with glioma (Fig. 7a, b).

Recent transcriptomic studies have shown that high-grade glio-mas (including glioblastoma, GBM) recapitulate early developmental states[11,50]. To explore this association with higher cell type resolution and a wider neurodevelopmental window, we examined our germinal matrix dataset in the context of published Neftel et al. (2019) GBM data defined into four cell states: astrocyte-like (AC-like), OPC-like, neural progenitor cell-like (NPC-like), and mesenchymal-like (MES-like)[50] (Fig. 7c–f, Supplementary Fig. 10). First, we plotted scores for AC-like, OPC-like, NPC-like, and MES-like gene lists in our dataset, and found concordant results between our developmental cell type signatures and the previously defined meta-modules[50]. We find tRG to be a pre-dominantly central population. Astrocyte, radial glia (RG/AC and oRG), and ependymal populations are distributed along the AC-like axis; OPCs and preOLs along the OPC-like axis; nIPCs, excitatory neurons (CPNs, SPNs, and CRNs), and inhibitory neurons (interneurons and MSNs) along the nIPC-like axis; and microglia, choroid plexus, and BVCs along the MES-like axis (Fig. 7c, Supplementary Fig. 10a). gIPCs and mIPCs are seen both centrally and distributed along the AC-like and NPC-like axes (Fig. 7c, Supplementary Fig. 10a). While gIPC-A map exclusively to the AC-like state, gIPC-O map to both OPC and NPC-like

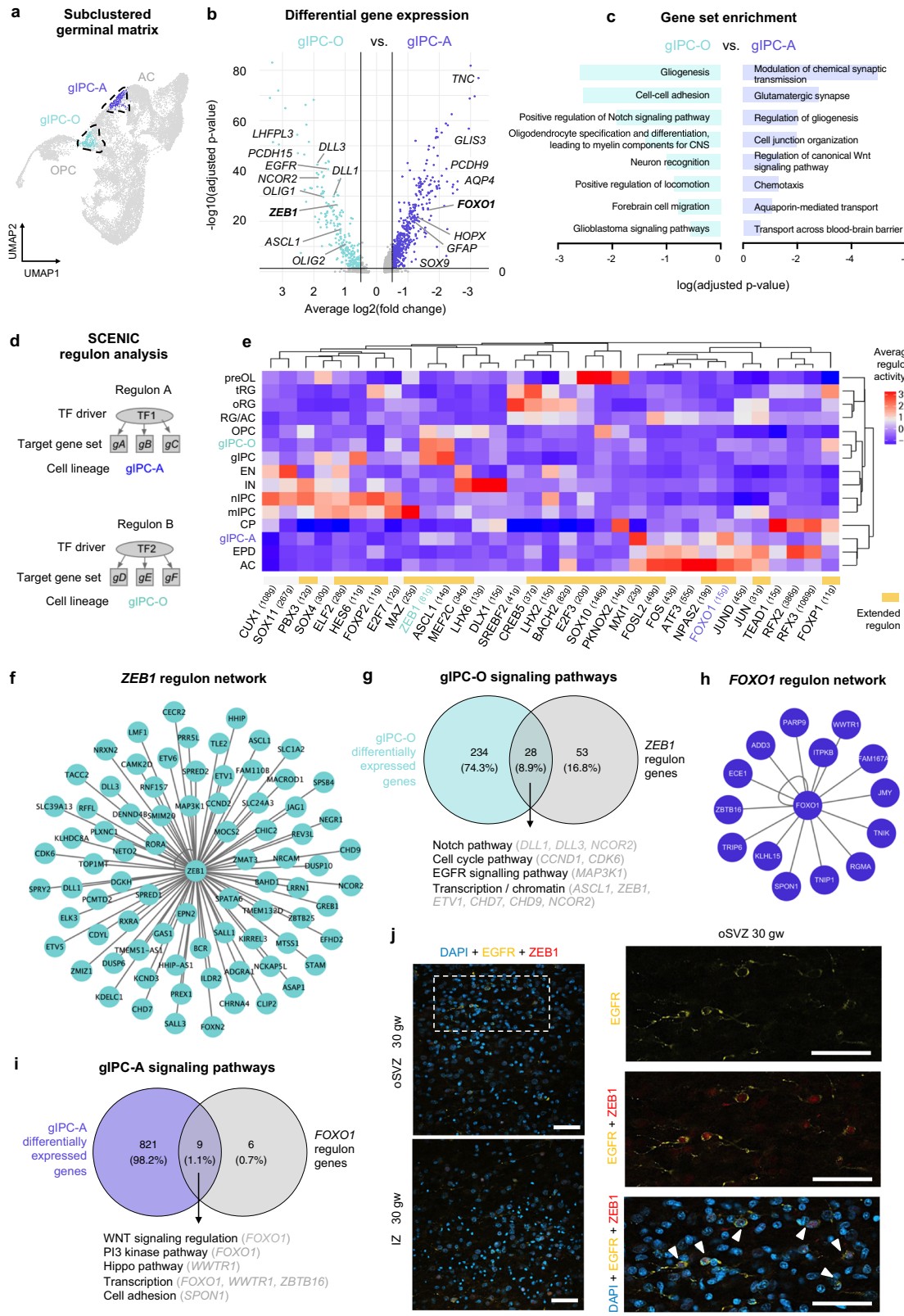

states. Of note, the NPC-like state can be subdivided into NPC-like state 1 (NPC1) based on the inclusion of oligodendrocyte lineage-related genes (*OLIG1*)[50], thereby justifying these distributions. Interestingly, this state is also defined by NOTCH pathway components (*DLL1* and *DLL3*), which we find to be associated with gIPC-O (Fig. 4g).

Next, we projected cell type-specific germinal matrix identities onto the malignant Neftel et al. (2019) single-cell dataset, composed of

pediatric and adult grade IV glial (GBM) tumors[50]. First, we calculated the enrichment scores for each prenatal signature in GBM cells (Fig. 7d) and then assigned the most similar prenatal signature to GBM (Fig. 7e, f). Interestingly, GBM samples tend to cluster according to developmental cell type similarity (Fig. 7e) rather than by patient identity (Supplementary Fig. 10b). Looking at the proportions of each prenatal signature calculated as the strongest match for each tumor

**Fig. 4 | Transcriptomic inference of regulatory biology in germinal matrix gIPC subtypes. a** UMAP plot of the subclustered germinal matrix analysis (from Fig. 2a) highlighting gIPC-O and gIPC-A subpopulations. **b** Volcano plot of DEGs between gIPC-O and gIPC-A (two-sided Wilcoxon Rank Sum test; p-adj. <0.05, average log2(fold change) > −0.5 or < −0.5). **c** Curated list of enriched terms computed from the top 100 gIPC-A (right) and gIPC-O (left) DEGs. Benjamini-Hochberg corrected p-adj <0.05. All terms are shown in Supplementary Data 10–11. **d** Schematic of the SCENIC regulon analysis. Regulons are composed of transcription factor regulators (TF1-2 on diagram) and their inferred gene targets (gA-F), namely co-expressed genes showing significant enrichment for the regulator's binding motif. **e** Heatmap of the average regulon activity z-score per cell subset. **f, h** Predicted ZEB1 (**f**) and FOXO1 (**h**) regulon networks from (**e**). **g, i** Venn diagrams quantifying the intersection of gIPC-O DEGs and *ZEB1* regulon genes (**g**) and of gIPC-A DEGs and *FOXO1* regulon genes (**i**). Below are biological processes and signaling pathways related to intersection genes. Source data for (**b, c, g**, and **i**) are provided as a Source Data file. **j** Single z-plane immunofluorescence images showing frequent colocalization of ZEB1 (red) with gIPC marker EGFR (yellow) (arrowheads) in oSVZ an IZ regions of a third trimester tissue sample (30gw). Images (right) show magnified inset of the area bordered in the oSVZ (top left). Immunofluorescence experiments in (**j–k**) were repeated three independent times, with similar results. Scale bar = 50 µm. RG radial glia, oRG outer radial glia, tRG truncated radial glia, EPD ependymal cell, AC astrocyte, gIPC glial intermediate progenitor cell, OPC oligodendrocyte progenitor cell, preOL premyelinating/early myelinating *BCAS1*+ oligodendrocyte, nIPC neuronal intermediate progenitor cell, mIPC multipotent intermediate progenitor cell EN excitatory neuron, CPN cortical projection neuron, SPN subplate neuron, CRN Cajal Retzius cell, IN interneuron, MG microglia, CP choroid plexus, oSVZ outer subventricular zone, IZ intermediate zone, TF transcription factor, gw gestational weeks.

cell (Fig. 7f), we find expected heterogeneity with the prominent representation of OPC, gIPC, mIPC, and astrocyte-like prenatal states. The prenatal mIPC signature, inferred as proliferative in our analyses, is consistently enriched in GBM cells, across all states. The gIPC-O signature preferentially matches to >50% of malignant cells in the GBM NPC1-like state and the gIPC-A signature is prevalent in the GBM AC-like state (Fig. 7f). NPC2-like, the state defined by neuronal lineage-related genes (RBFOX1/2, DLX6-AS1, and DLX5)[50], is primarily interneuronal. The MES1-like and MES2-like states, respectively defined by hypoxia-independent and -dependent programs tumors[50], are both primarily astrocytic. This is consistent with the inclusion of astrocyte markers (*VIM* and *CD44*) in their meta-modules. Overall, this analysis identifies gIPC-A and gIPC-O developmental states as additional contributors to glioblastoma heterogeneity and provides high-resolution signatures for further study in the context of human disorders.

## Discussion

We have generated a comprehensive transcriptomic atlas of second and third-trimester human neocortical development, capturing the diversity of cell types, states, and differentiation trajectories in the germinal matrix and cortical plate. This unique snRNA-seq dataset is a necessary complement to available data from earlier prenatal stages[1–4,7,12–16,19,51–54]. It greatly expands our understanding of third-trimester neurodevelopment, a critical period of gliogenesis, by uncovering distinct yet developmentally transient lineages with links to specific pathological states[8]. Importantly, our computational analyses resolve both oligodendrocyte and astrocyte lineages up until the time of birth and infer human gIPC populations with distinct lineage bias towards oligodendrocyte progenitors (gIPC-O), and towards astrocytes (gIPC-A), two cellular state identities that we find to be represented in adult and pediatric glioblastoma tumors. This dataset is publically available to facilitate further interrogation of neocortical cell types within late prenatal development, including their potential roles in pediatric and adult-onset disorders.

The generalizability of our data ultimately relies on independent validation. To this end, we find high concordance in the cell types identified between our mid-gestation dataset and published datasets of the same[3,12,15,16]. Multipotent progenitors were first identified in the embryonic mouse neocortex as a proliferative EGFR+/ASCL1+/OLIG1+/OLIG2+ cell type that gives rise to astrocyte, oligodendrocyte, and olfactory bulb interneurons[19]; more recently their existence in the prenatal human neocortex at mid-gestation was shown as well[12]. While we find a corresponding EGFR+/ASCL1+ mIPC population in the germinal matrix, our dataset clearly resolves a separate glial progenitor population during late prenatal gestation within the germinal matrix and the cortical plate, an EGFR+/ASCL1+ and OLIG1+/OLIG2+ gIPC with trajectory towards glial differentiation only. Notably, mIPC clusters emerge after cell cycle regression and show differential upregulation of several proliferation markers compared to gIPCs. This may explain the correspondence between mIPCs and the cycling cell

population defined by Fu et al. (2021)[16]. Moreover, while gIPC and nIPC populations form distinct clusters in our lineage trajectories analyses, mIPCs do not. Monocle analysis shows mIPCs populating multiple differentiation trajectories, and inference of lineage directionality through scVelo highlights mIPCs as a heterogeneous population that precedes many others, including gIPC, excitatory neuron (via nIPC), interneuron, oRG (via RG/ACs), astrocyte (directly and indirectly via gIPC-A or RG/ACs), and oligodendrocyte (via gIPC-O). These observations suggest that mIPCs may represent a cell state rather than a lineage-defined cell type, a notion put forward in several prior studies[12,16,19,40,55]. Truncated RG, which are similarly multipotent[2,4,56,57], consistently cluster with mIPCs in our trajectory analyses, and overall represent the most likely origin for gIPCs. Of note, we have previously seen[40] and herein confirm enrichment of EGFR+ cells in the VZ/iSVZ at 20 gw, where tRG reside. We have also shown that EGFR+ cells acutely isolated from postmortem human germinal matrix tissue at midgestation display stem cell-like properties in vitro, including self-renewal and trilineage differentiation potential[40]. We now speculate that this EGFR+ stem-like population[40] likely represents bMIPCs identified by Yang et al.[12] and mIPCs annotated in this study.

In contrast, our computational and histological studies put forward "gIPC" as a glial-specific progenitor cell type, transiently present during neurodevelopment. At 30gw, we find EGFR+ cells to be largely absent from the VZ/iSVZ with many EGFR+/OLIG2+ cells seen instead in the oSVZ and intermediate zone, coinciding with a decrease in mIPC proportions and presumed outward migration of gIPC in the third trimester. The characteristics of the herein-defined late prenatal human gIPC are consistent with a previous report of EGFR+ bipotent glial progenitors in the oSVZ of the non-human primate neocortex at late gestation[10] in addition to other animal models of neurodevelopment[9,10,18,20,58–61]. The resolution in our dataset enabled us to further define gIPC subpopulations with distinct bias towards oligodendroglial and astrocyte identity, namely EGFR+/OLIG2+/SOX9- gIPC-O and EGFR+/SOX9+/OLIG2- gIPC-A, and to infer biological pathways and regulatory drivers of each intermediate. Regulon analysis prioritized ZEB1 as a putative TF regulator in gIPC-O biology. We further validated ZEB1 expression in gIPCs in situ, and speculate that it may operate through NOTCH signaling, a pathway recently implicated in OPC development[54]. While our multipronged approach to trajectory inference provides robust evidence for gIPC lineage bifurcation, all lineage inferences in this study are purely computational and must be ultimately validated in human models, such as through ex vivo lineage tracing.

Finally, we leveraged this prenatal dataset as a unique, transient snapshot into neurodevelopment and gliogenesis, to discover abundant representation of gIPC-A and gIPC-O developmental states in glioblastoma heterogeneity, across adult and pediatric tumors. While the idea of a common glial progenitor as the GBM cell of origin has been proposed elsewhere[11,25,34], our analysis contributes to the refinement of developmental cell state signatures uniquely derived from

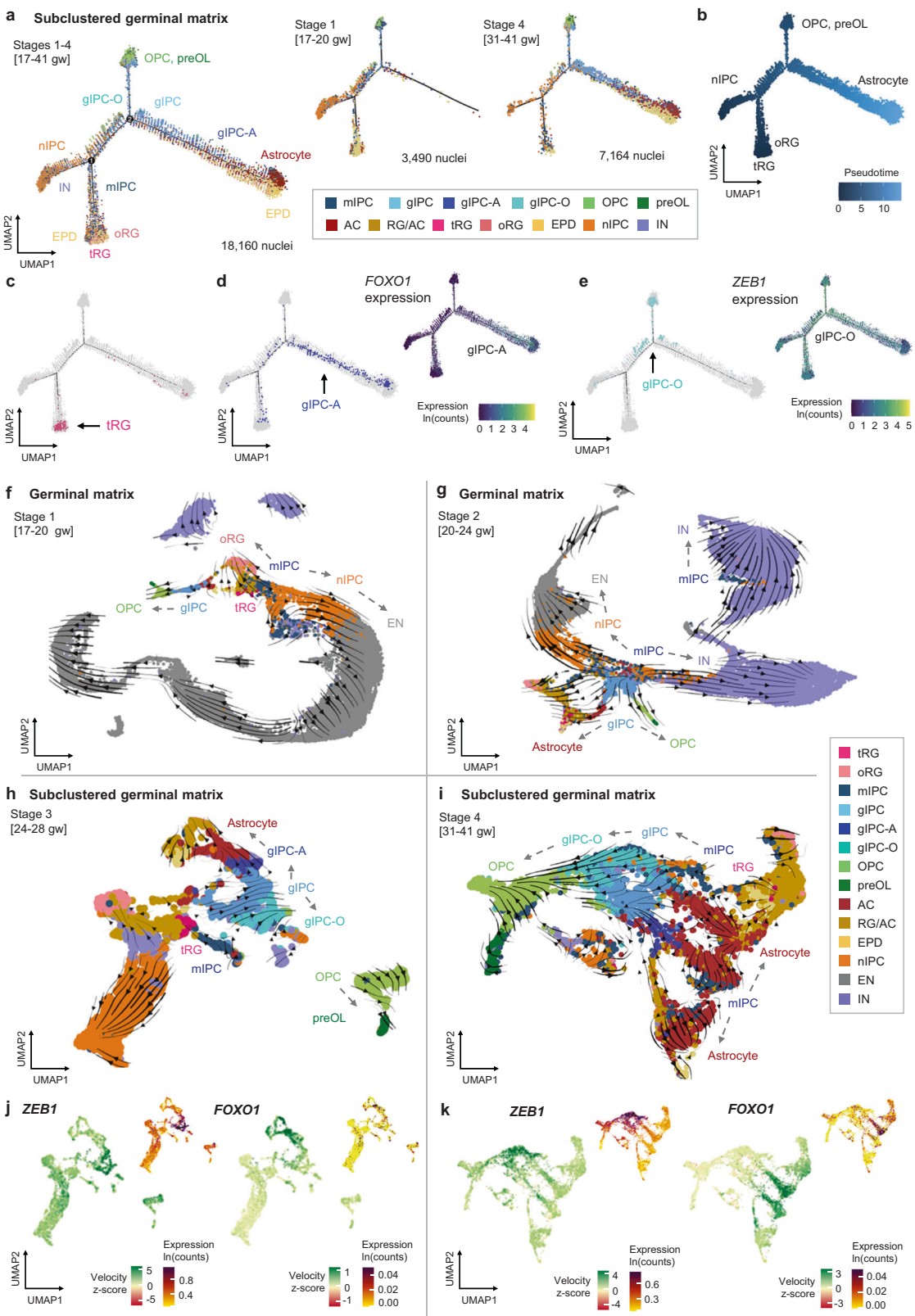

prenatal human germinal niches and demonstrates how our dataset can be utilized in the study of human pathologies. Gliomas co-opt developmental pathways to maintain cell proliferation and migration, properties associated with EGFR activation[40,49,62,63]. While cell type correspondence analyses must be interpreted with caution, particularly in the study of heterogeneous neoplasms such as glioblastoma, we speculate that further studies into gIPC lineage-specific TFs and

their regulatory networks may facilitate the discovery of novel therapeutic targets in this otherwise incurable disease.

## Methods

### Sample collection

All specimen collection was performed on postmortem samples, resulting from spontaneous loss not attributable to known prenatal

**Fig. 5 | Reconstruction of cell type lineages in the germinal matrix.**
**a**–**e** Trajectory reconstruction of GM lineages (subclustered cell type annotations) using Monocle2, showing the distribution of different cell subsets (**a**, **c**–**e**) and inferred pseudotime (**b**). tRGs were selected as the root population. Average log-normalized gene expression of the gIPC-A marker *FOXO1* (**d**, right), and the gIPC-O marker *ZEB1* (**e**, right) are also shown. **f**–**i** Directionality analysis of GM cell types. Velocity vectors (solid arrows) calculated for GM Stages 1–4 (17–41 gw) using stochastic modeling in scVelo, and projected onto stage-specific UMAP embeddings. Cells are colored by cluster annotations defined in Fig. 1b (**f**–**g**) and Fig. 2a (**h**–**i**). Dashed arrows indicate putative lineages. **j**–**k** UMAP plots showing the velocity z-scores and log-normalized expression of *ZEB1* (left) and *FOXO1* (right) along respective gIPC-O and gIPC-A lineage trajectories, in stage 3 (**j**) and stage 4 (**k**). Color scale in velocity plots corresponds to transcriptional induction (green) and repression or absence of transcription (red), inferred from the ratio of unspliced to spliced mRNA. See also Supplementary Figs. 7 and 8. RG radial glia, oRG outer radial glia, tRG truncated radial glia, EPD ependymal cell, AC astrocyte, gIPC glial intermediate progenitor cell, OPC oligodendrocyte progenitor cell, preOL pre-myelinating/early myelinating *BCAS1*+ oligodendrocyte, nIPC neuronal intermediate progenitor cell, mIPC multipotent intermediate progenitor cell CPN cortical projection neuron, IN interneuron.

abnormalities. Sample collection was performed de-identified, under appropriate consent, and in accordance with the policies and regulations at the Icahn School of Medicine at Mount Sinai and its institutional review board. One hemisphere was sectioned coronally fresh (unfixed) and then snap-frozen at −80 °C in a dedicated banking area. The other hemisphere was fixed in 10% formalin for ~1 week, and used for in situ validation as well as to confirm histologically the lack of diagnostic neuropathological abnormalities. The gestational age of samples was calculated based on three independent metrics: (1) clinical estimation based on the last menstrual cycle, (2) foot length, and (3) brain weight[64]; sex was confirmed using *XIST* gene expression.

### Tissue sample dissection and nuclei isolation

All samples were processed over a three-week period, with samples across age groups and regions processed in parallel on any given experiment day, to minimize batch effects. Samples were slowly semi-thawed to −20 °C and processed on ice to maintain gross anatomical structure. The germinal matrix was macrodissected at the level of the caudothalamic groove and the cortical plate from the adjacent frontoparietal neocortex; grossly equal amount of subjacent white matter was included for both dissections. To consistently dissect the posterior germinal zone in prenatal brains of varying size, the length of the entire cortical surface was measured for each sample and tissue from the ventricular to the apical surface was dissected in the area three fourths of the length from the rostral aspect. Histological confirmation of the dissection was performed by hematoxylin-eosin staining for most samples using adjacent tissue. For adult tissues, the neocortex (CX) was dissected at the level of the pre-central gyrus, Brodmann area 4, and the subventricular zone (SVZ) plus caudate were dissected at the level of the posterior basal ganglia adjacent to the dissected CX.

Tissues were dissociated by douncing 50–100 mg of prenatal tissue or 250 mg of adult tissue in 4 ml of lysis buffer (0.32 M sucrose, 5 mM calcium chloride, 0.1% Triton X-100, 0.1 mM ethylenediaminetetraacetic acid [EDTA], 3 mM magnesium acetate, and 1 mM dithiothreitol [DTT] in 1 mM Tris-HCl, pH 8.0) under RNase-free working conditions. Nuclei were isolated by sucrose gradient ultracentrifugation[65]. Briefly, a sucrose solution (1.8 M sucrose, 3 mM magnesium acetate, and 1 mM DTT in 10 mM Tris-HCl, pH 8.0) was layered under the lysate solution followed by centrifugation at 4 °C for 1 h at 102,000 x g. Nuclei were resuspended in 0.04% bovine serum albumin (BSA) in Ca2+/Mg2+ -free phosphate buffered saline (PBS) at $1 \times 10^6$ cells/mL concentration and loaded to achieve optimal recovery of 6000 sequenced nuclei/sample with minimal doublet/triplet formation. Concentrations were verified using an automated cell counter (Countless 3, ThermoFisher Scientific). Trypan blue staining was used to assess nuclei quality and quantity prior to library preparation.

### RNA library preparation for high-throughput single-nucleus sequencing

RNA from single nuclei was prepared for sequencing using the 10X Genomics Chromium platform and the 3′ gene expression (3′ GEX) V3

kit, version 3.10. Each region (GM, CP, SVZ, CX) and the sample was barcoded and sequenced as separate libraries, without hashing. Nuclei were diluted and loaded with a target of 6,000 cells into nanoliter-scale Gel Bead-In-Emulsions (GEMs). Primers containing Illumina read 1 (R1) sequencing primers, a 16-bp 10x Barcode, a 10 bp randomer and a poly-dT primer sequence were subsequently mixed with the nuclear suspension and master mix. After incubation of the GEMs, barcoded, full-length cDNA was generated from pre-mRNA. Then, the GEMs were broken and silane magnetic beads were used to remove leftover biochemical reagents and primers. Prior to library construction, enzymatic fragmentation and size selection were used to optimize cDNA amplicon size. P5 and P7 primers, a sample index, and Illumina read 2 (R2) sequencing primers were added to each selected cDNA during end repair and adaptor ligation. P5 and P7 primers were used for Illumina bridge cDNA amplification (http://10xgenomics.com). Libraries were quantified using the Agilent Bioanalyzer and sequenced into 2 × 100 paired-end reads using the Illumina NovaSeq platform to target 50,000–100,000 reads per nucleus. Cell Ranger (10X Genomics) was used to demultiplex reads and count unique transcripts of Ensemble genes with default parameters (v2.0.1) by mapping to the GRCh38 human genome pre-mRNA transcriptome sequences; such mapping enables comparable cluster resolution to single-cell RNA sequencing (scRNA-seq)[66].

### Data integration and clustering

Gene expression count matrices were generated with CellRanger (v3.1.5), processed and analyzed mainly using R package Seurat[67] (v4.0.4). Doublet rates were estimated per sample based on the number of nuclei recovered (https://satijalab.org/costpercell) and using the R package, DoubletFinder[68] (v2.0.3). Low quality nuclei (<400 unique genes, <1000 UMI counts, or >15% mitochondrial genes) and doublets were excluded from downstream analyses. When analyzing individual samples, gene expression for each nucleus was normalized by the total counts to adjust for the sequencing depth, multiplied by a scale factor of $10^4$, and log-transformed (*NormalizeData* function in Seurat). Dimensionality reduction was applied on the scaled expression of the top 2000 variable genes by Principal component analysis (PCA) (Seurat::*RunPCA*). To mitigate inter-sample technical variation when analyzing multiple samples, we normalized and integrated the data using the *SCTransform* workflow in Seurat, which finds mutual nearest neighbors[69] (or anchors) to integrate different samples. PCA was recomputed using the top 3000 variable features followed by nearest-neighbor graph construction (Seurat::*FindNeighbors*), unbiased Louvain clustering (Seurat::*FindClusters*), and Uniform Manifold Approximation and Projection (UMAP)[30] (Seurat::*RunUMAP*) for visualization of the data in two dimensions.

### Data visualization

Data were visualized using R packages ComplexHeatmap[70](v2.8.0), ggplot2(v3.3.3), ggrepel (v0.9.1), ggvenn (v0.1.9), and built-in Seurat functions, in addition to GraphPad Prism 9 (v9.3.1 for macOS) and Cytoscape[71] (v3.9.1 for macOS).

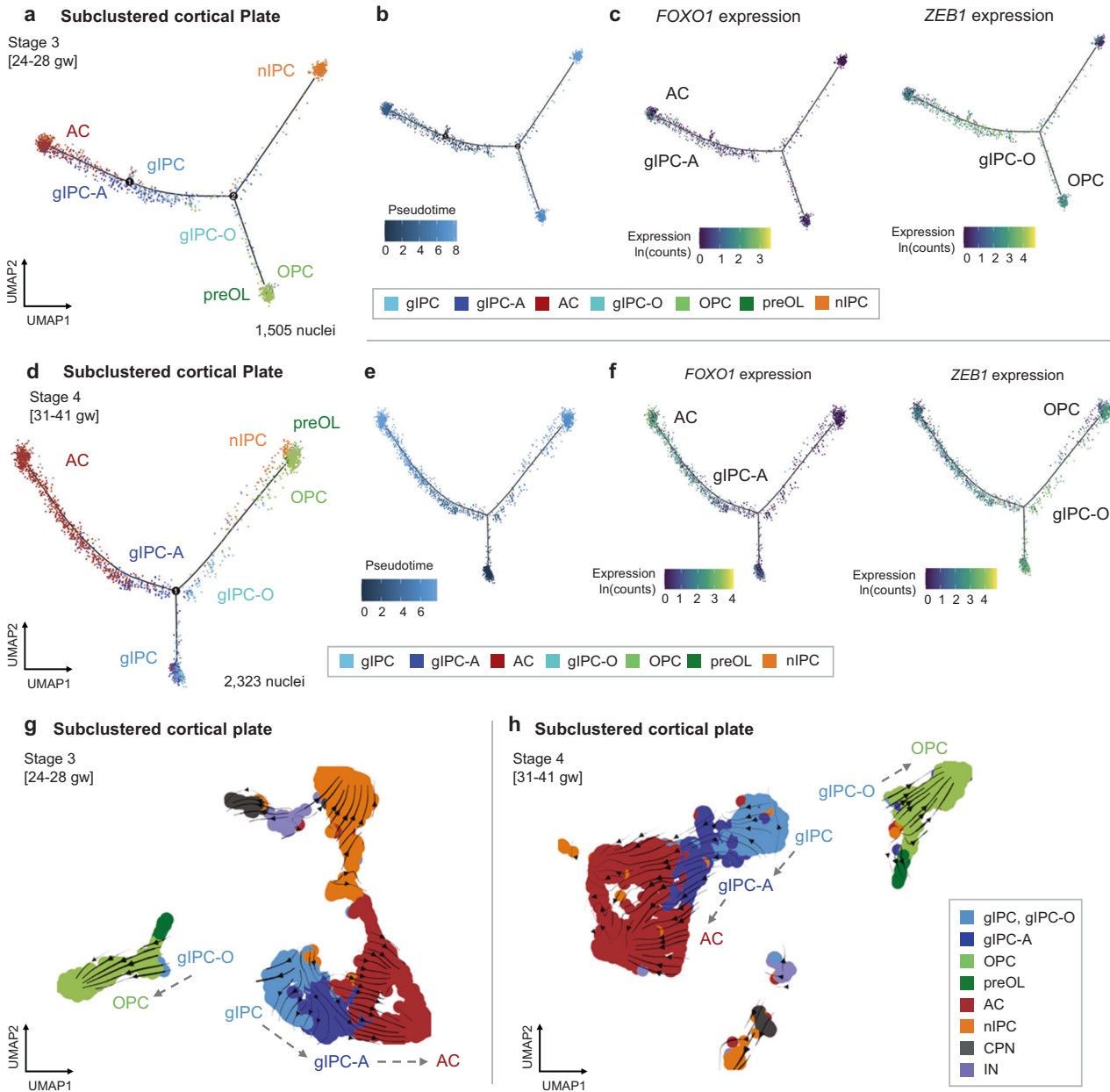

**Fig. 6 | Reconstruction of cell subtype lineages in the cortical plate.**
**a–f** Trajectory analysis of CP lineages (subclustered cell type annotations from Fig. 2h) for stages 3 (**a**–**c**) and 4 (**d**–**f**) using Monocle2. Pseudotime (**b**, **e**) and the average log-normalized gene expression of *FOXO1* (**c**, **f**; left), and *ZEB1* (**c**, **f**; right) are also shown. gIPCs are selected as the root population. **g**–**h** Directionality analysis of CP cell types. Velocity vectors (solid arrows) calculated for stages 3 (**g**) and 4 (**h**) using stochastic modeling in scVelo, projected onto stage-specific UMAP embeddings. Cells are colored by cell subtype annotations from Fig. 2h. See also Supplementary Fig. 8. AC astrocyte, gIPC glial intermediate progenitor cell, OPC oligodendrocyte progenitor cell, preOL premyelinating/early myelinating *BCAS1+* oligodendrocyte, nIPC neuronal intermediate progenitor cell, CPN cortical projection neuron IN interneuron, CP prenatal cortical plate, gw gestational weeks.

## Canonical marker analysis and de novo marker identification

Cell cycle scoring was performed using established cell cycle phase markers[38] and nuclei were classified into G2/M-, S-, and G1-phase (Seurat::*CellCycleScoring*). Nuclei with G2/M- or S-phase scores of >0.25 were annotated as cycling or transient amplifying cells (TAC). G2/M-, S-phase scores were regressed in subclustered analyses for the purpose of removing the effect of the cell cycle on lineage trajectory inference. Other nuclei were annotated using canonical marker gene expression averaged per cluster. Differential gene expression analysis was performed to identify statistically significant genes using non-parametric two-sided Wilcoxon Rank Sum testing (Seurat::*FindAllMarkers*). Gene set enrichment analyses were performed using Metascape[72] by selecting the top 100 differentially expressed

genes (DEGs) ranked by average log2(fold change) and FDR adjusted *p*-value <0.05.

## Lineage enrichment and maturation analyses

Lineage enrichment and maturation scores were computed using the top 50 DEGs for different cell types (e.g., astrocyte, oligodendrocyte) (Seurat::*FindAllMarkers*, Seurat::*AddModuleScore*), ranked by average log2(fold change) and FDR adjusted *p*-value <0.05 for each population (*wilcox.test*). Lineage enrichment of astrocyte, OPC, interneuron, and nIPC fate was computed using DEGs derived from the cell cycle-regressed subclustered data. Maturation was studied in unregressed subclustering analysis based on astrocyte, oligodendrocyte, excitatory neuron, and interneuron DEGs.

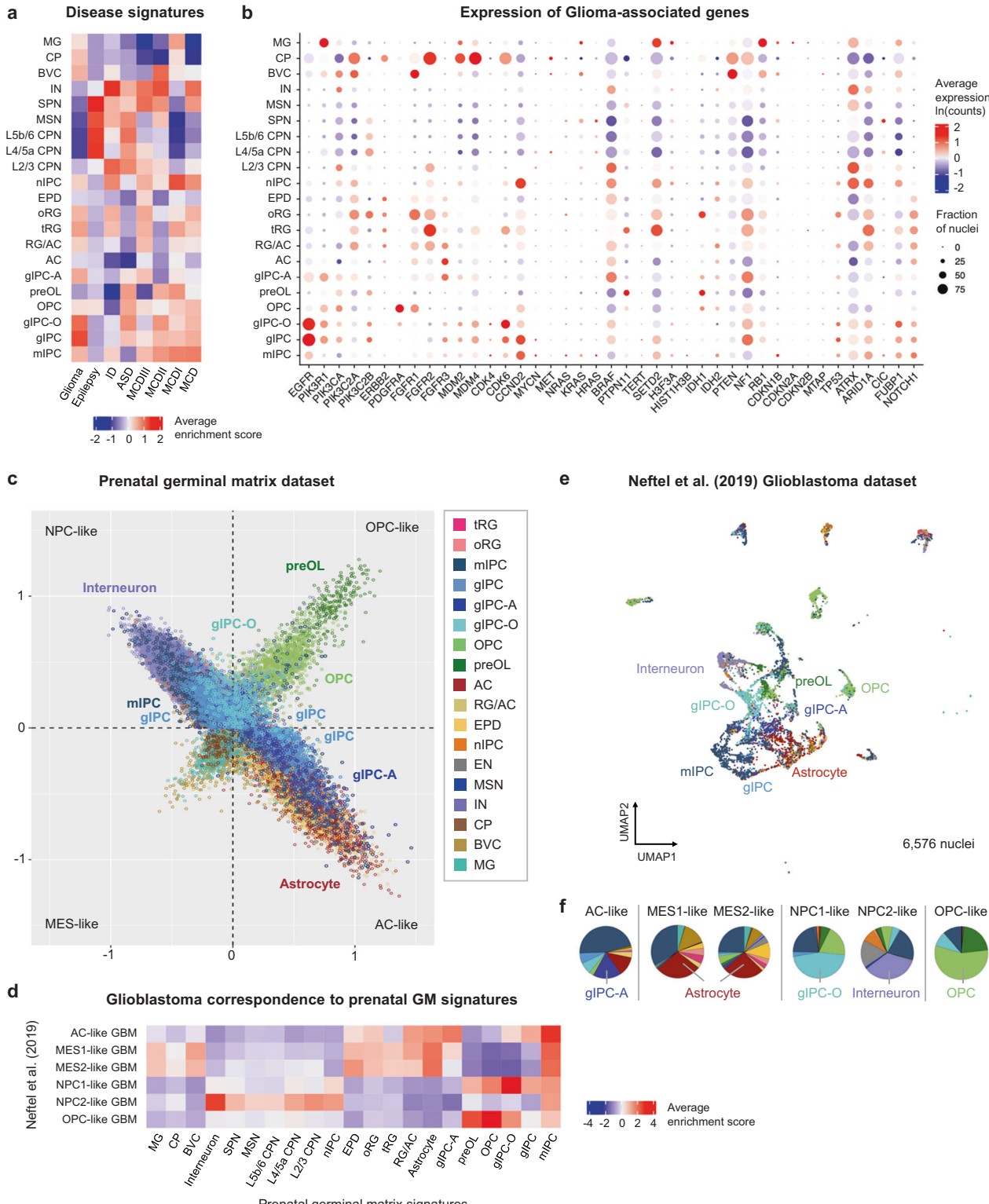

## Correspondence analysis

For prenatal/adult and prenatal/published dataset correspondence analyses, we compared de novo annotations based on unsupervised clustering to annotations derived based on the label transfer workflow in Seurat (Seurat::*FindTransferAnchors* and Seurat::*TransferData*) from an annotated reference. To compare de novo prenatal annotations in this study to other published studies, we used annotations from published studies as references. Count matrix of published datasets with annotations was provided generously from the authors upon our request. To assess if prenatal cell types are found in our adult data, we used the same workflow but now used our prenatal annotations as the reference and measured their correspondence to adult annotations. For visualization, confusion matrices were generated, quantifying the correspondence between our de novo annotations and

**Fig. 7 | Mapping of prenatal cell type signatures to human disorders. a** Heatmap showing z-scored enrichment for module scores calculated from published/curated lists of disease-associated genes. Z-scores are averaged per cell subtype. **b** Dot plot showing log-transformed expression averaged per cell subtype and proportion of nuclei that express each glioma-associated gene. **c** Scatter plot representation of enrichment scores for prenatal germinal matrix cell types calculated using published signatures used to define glioma cell states ("MES-like", "AC-like", "NPC-like", "OPC-like")[50]. **d** Heatmap showing the average enrichment scores of all GM cell type signatures assigned to neoplastic cells from an external single-cell RNA-seq dataset of adult and pediatric grade IV gliomas (glioblastomas)[50]. Average enrichment scores are computed using the top 50 DEGs for each GM cell type (DEG significance calculated using two-sided Wilcoxon Rank Sum test with p-adj. > 0.05; average log2(fold change) >0.5 or <−0.5). **e** UMAP plot of GM cell type assignments to GBM

cells in the Neftel et al dataset[50], based on highest enrichment score from (**d**). **f** Pie charts quantifying the proportions of GM cell types assigned onto GBM cells from (**e**), divided into glioma cell states ("OPC-like","AC-like", "NPC1/2-like", and "MES1/2-like"). The predominant predicted cell type for each state is included below. Source data for (**f**) are provided as a Source Data file. See also Supplementary Figs. 9 and 10. RG radial glia, oRG outer radial glia, tRG truncated radial glia, EPD ependymal cell, AC astrocyte, gIPC glial intermediate progenitor cell, OPC oligodendrocyte progenitor cell, preOL premyelinating/early myelinating *BCAS1*+ oligodendrocyte, nIPC neuronal intermediate progenitor cell, mIPC multipotent intermediate progenitor cell MSN medium spiny neuron, EN excitatory neuron, CPN cortical projection neuron, SPN subplate neuron, CRN Cajal Retzius cell, IN interneuron, MG microglia, BVC blood vessel cell, L2/3, L4, L5/6 neocortical layers 2/3, 4, 5/6, GM prenatal germinal matrix, CP choroid plexus, gw gestational weeks.

predicted labels when using another annotated data set as a reference. Cells with prediction scores of <0.5 were deemed unreliable and annotated as unclassified.

## Neurodevelopmental disorder and disease analyses

A curated list of glioma-associated genes[49] and published lists of neurodevelopmental disorder-associated genes[3,47,48] were used to calculate disease enrichment scores per cell type (Seurat::*AddModuleScore*). For the reanalysis of published GBM data[50], gene expression for each cell was normalized by the total cell counts, multiplied by a scale factor of $10^4$, and log-transformed (Seurat::*NormalizeData*). Dimensionality reduction was applied on the scaled expression of top 2000 variable genes by PCA (Seurat::*RunPCA*) followed by nearest-neighbor graph construction (Seurat::*FindNeighbors*), Louvain clustering (Seurat::*FindClusters*), and UMAP analyses (*runUMAP*). Mapping of prenatal annotations (this study) to GBM malignant cells (Neftel et al, 2019) was performed by computing enrichment scores for each prenatal cell type signature (Seurat::*FindAllMarkers*, Seurat::*AddModuleScore*), and then assigning the highest enrichment score signature to each GBM cell. Enrichment scores were calculated for each prenatal cell type, using its top 50 DEGs ranked by average log2(fold change) and FDR adjusted *p*-value.

## Cell type proportions

Cell type proportions were calculated as a percentage of the total number of nuclei per anatomical region, excluding nuclei that were classified as 'Undefined' (UD). The proportions of cell types within the TAC population were calculated as a percentage of the total number of TACs. Scatter plots displaying the correlation between cell type proportion (percentage of total nuclei) and gestational weeks across samples were fitted with a linear model (*lm*) using the R package, stats (v3.6.2). Data normality was tested by Shapiro-Wilk normality testing (stats::*shapiro.test*), and statistical significance for cell type proportions (*p*-value <0.05) was calculated by two-tailed Spearman's Rank-Order Correlation analysis (*cor.test*). The robustness of these results were assessed by leave-one-out cross-validation, and confidence intervals for the correlation and p-values were reported.

## Regulon analysis

Regulon analysis was performed using the R package, SCENIC[73] (v1.1.2−01), using default parameters. Regulon activity was averaged across cells of the same annotation and displayed in a heatmap, showing the most highly enriched regulons per annotation.

## Lineage trajectory reconstruction and directionality analyses

Developmental trajectories were reconstructed using several lineage inference methods. Diffusion maps were computed with Euclidian distance and local diffusion scale using the R package, destiny[43]. The R package, Monocle2[42] (v2.20.0), was used to map complex branching trajectories in pseudotime (Monocle::*plot_cell_trajectory*). Differential

genes were calculated by cell type (*differentialGeneTest*) and the top 3000 genes (FDR adjusted p-value <1e$^{-3}$) were selected for ordering (*setOrderingFilter*). Dimensionality reduction was performed using the DDRTree package (version 0.1.5) algorithm (*reduceDimension*). Nuclei were ordered based on the root state that included tRG (*orderCells*). Gene expression was plotted using log-transformed expression data from the corresponding Seurat object (*plot_cell_trajectory*). RNA velocity and trajectory direction was inferred per stage using the Python package, scVelo[44,46] (version 0.2.4). Sample-specific loom files were generated using the R package, velocyto[45] (v0.6), and merged into stages using the Python package, Loompy (v3.0.0). Velocities were calculated using a stochastic model (*scv.tl.velocity*) and projected as streamlines (*scv.pl.velocity_embedding_stream*) onto a pre-computed UMAP embedding (Seurat::*RunUMAP*). Genes of interest or genes identified by differential velocity expression between cell types (*scv.tl.rank_velocity_genes*) were visualized as phase portraits (*scv.pl.velocity*).

## Immunofluorescence

Formalin-fixed, paraffin-embedded slides of human prenatal brains were incubated at 70 °C for 1 h min followed by two 10-min washes in 100% xylene and one 10-min wash in 50% xylene 50% ethanol. Tissues were re-hydrated by graded alcohol washes (5 min in 100%, 95%, 75%, 50%, and 25% ethanol in Milli-Q water) followed by two 10-min washes in Milli-Q water. They were subsequently treated in a citrate-based antigen unmasking solution, pH 6.0 (Vector Laboratories, H-3300-250), at 92 °C for 45 min, cooled to room temperature, and washed three times for 5 min in Milli-Q water. Paraformaldehyde-fixed, OCT-embedded human prenatal brain tissues were cryosectioned at 50 μm thickness and washed three times for 10 min in 1 X PBS. Floating sections were incubated in a citrate-based antigen unmasking solution, pH 6.0, at 70 °C for 2 h, cooled to room temperature and washed three times for 10 min in 1X Tris-buffered Saline (TBS). Mounted or floating sections were blocked with 10% donkey serum (DS) in 1X TBS with 0.1% Triton X100 (0.1% TBS-TX) for 1 h at room temperature and incubated with primary antibodies (1:50 goat anti-EGFR, R&D Systems AF231; 1:50 mouse anti-EGFR, Abcam ab218383; 1:200 rabbit anti-SOX9, Abcam ab185966; 1:50 rabbit anti-OLIG2, Millipore AB9610; 1:200 goat anti-OLIG2, R&D Systems AF2418; 1:500 rabbit anti-ZEB1, Invitrogen PA5-82982; 1:50 rabbit anti-Ki67, Invitrogen PA1-38032) in 1% DS in 0.1% TBS-TX overnight at 4 °C. Following four 15-min washes in 1X PBS for 15 min, they were incubated with secondary antibodies (1:500 donkey anti-rabbit Alexa Fluor 488, Jackson ImmunoResearch Labs 711545152; 1:500 donkey anti-goat Alexa Fluor 594, Jackson ImmunoResearch Labs NC0281835) in 1% DS in 0.1% TBS-TX for 2 h at room temperature. Slides were washed four times for 15 min in 0.1% TBS-TX and incubated for 10 min with 300 nM DAPI (Invitrogen D1306) in 0.1% TBS-X. They were then washed for another 5 min in 0.1% TBS-TX and mounted with Aqua-Poly/Mount (Polysciences, 18606-20) before storing at 4 °C. Images were captured using a Zeiss LSM 780 upright microscope and processed using Zen Black software and ImageJ (version 2.3.0/1.53q).

**Reporting summary**

Further information on research design is available in the Nature Portfolio Reporting Summary linked to this article.

## Data availability

Single nucleus prenatal and adult RNA-seq data generated in this study have been deposited in the GEO database under accession code GSE217511. Previously published data used in this study for comparative studies are available in the GEO and GSA databases under accession codes GSE131928 (Neftel et al 2019, GBM); HRA000348 and GSE144462 (Yang et al 2022, Fu et al 2021; human development), and GSE161132 (Li et al 2021; mouse development). Source data are provided with this paper.

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

## Acknowledgements

We would like to acknowledge the late Dr. Mary Fowkes, whose passion for diagnostics and research in neurodevelopment inspired this study, as well as fellows and staff under her leadership who contributed to tissue procurement. This work was supported in part through the computa-tional resources and staff expertize provided by Scientific Computing at the Icahn School of Medicine at Mount Sinai.

## Author contributions

Conception of project: N.M.T. Tissue procurement: N.M.T. and F.D. Experimental design: N.M.T. and E.F (biological), A.M.T. (computational). snRNA-seq data preparation: R.S., K.B., E.F., K.A., and Z.M. Bioinformatic analysis: S.I.R., Z.M., B.G., P.C., and A.M.T. Data interpretation: Z.M., S.I.R., E.F., A.M.T., and N.M.T. Histology: S.I.R. and B.P. Manuscript pre-paration: S.I.R., Z.M., N.M.T., E.F., and A.M.T. Editing: all authors. This research was supported by RF1DA048810, R61DA048207, and R01NS106229 (N.M.T.).

## Competing interests

The authors declare no competing interests.
