## [Peer Review File · Nature Communications]

An atlas of late prenatal human neurodevelopment resolved by single-nucleus transcriptomicsEditorial Note: This manuscript has been previously reviewed at another journal that is not operating a transparent peer review scheme. This document only contains reviewer comments and rebuttal letters for versions considered at *Nature Communications*. Mentions of the other journal have been redacted.

REVIEWERS' COMMENTS

Reviewer #1 (Remarks to the Author):

The authors responded adequately to my comments and I support the publication of this manuscript.

Reviewer #4 (Remarks to the Author):

This atlas of neural development by snRNAseq will serve as an important resource for the community, and that alone justifies publication in Nat Comm. While the study is descriptive in nature, the dataset will serve as important resource for the community. It also appears that the authors have had to address prior comments from [redacted] reviewers. This reviewer finds their answers to the comments adequate and will avoid adding too many additional comments.

2 minor comments that the authors could address:

- trajectories in single-cell data, while widely used, have very limited validations; approaches were developed primarily to resolve trajectories in normal samples over a few hours rather than over many months of dev; and such approaches are highly dependent on defining a root etc and assume trajectories even when none is biologically present; at minimum, the authors should discuss the limitations of such inferred trajectories.
- transcriptional similarities to disease are informative but can be very misleading; cancer is a very distorted version of development and precisely assigning a given cell type/state is a very complex (and risky!) procedure that requires a lot of caveat; the authors should discuss those points and remove strict annotations as done in their pie charts in fig 7E; these annotations should reflect a degree of similarity rather than a strict/precise assignment; a score-based approach as done in 7C is preferable (Jaccard indexes could be alternative way of displaying the data).

REVIEWERS' COMMENTS

Reviewer #1 (Remarks to the Author):

The authors responded adequately to my comments and I support the publication of this manuscript.

We thank the reviewer for taking the time to review our work for a second time and for the original suggestions made, addressing which greatly improved our manuscript. We are pleased that our revised work has adequately addressed their concerns.

Reviewer #4 (Remarks to the Author):

This atlas of neural development by snRNAseq will serve as an important resource for the community, and that alone justifies publication in Nat Comm. While the study is descriptive in nature, the dataset will serve as important resource for the community. It also appears that the authors have had to address prior comments from [redacted] reviewers. This reviewer finds their answers to the comments adequate and will avoid adding too many additional comments. 2 minor comments that the authors could address:

We thank the reviewer for highlighting the value of our dataset and for their comments listed below, which we have now addressed.

- trajectories in single-cell data, while widely used, have very limited validations; approaches were developed primarily to resolve trajectories in normal samples over a few hours rather than over many months of dev; and such approaches are highly dependent on defining a root etc and assume trajectories even when none is biologically present; at minimum, the authors should discuss the limitations of such inferred trajectories.

We agree with the reviewer that computational inference of lineage trajectories has limitations. As suggested, we now acknowledge the limitations in the text (page 19) and discuss the need for lineage tracing to definitively establish lineage relationships discussed in the manuscript. As we recognize that each computational methodology has underlying assumptions and is imperfect, the proposed gIPC lineage in our study is supported by the orthogonal inference of lineage trajectories using three independent computational packages (Destiny diffusion maps, Monocle2, and ScVelo) in combination with extensive immunohistochemical validation in primary tissue samples. Furthermore, all interpretations in this study considered previously published data, including a study using a human slice culture model (Huang et al, *Cell* 2020) and our own *in vitro* differentiation model of prospectively isolated human EGFR+ germinal matrix progenitors (Tome-Garcia et al, *Stem Cell Reports* 2017).

- transcriptional similarities to disease are informative but can be very misleading; cancer is a very distorted version of development and precisely assigning a given cell type/state is a very complex (and risky!) procedure that requires a lot of caveat; the authors should discuss those points and remove strict annotations as done in their pie charts in fig 7E; these annotations should reflect a degree of similarity rather than a strict/precise assignment; a score-based approach as done in 7C is preferable (Jaccard indexes could be alternative way of displaying the data).

We agree with the reviewer that analyzing developmental signatures in the context of heterogenous disease, such as glioblastoma, is complex and have followed their guidance to

discuss caveats in our discussion (page 19-20) as well as to improve on our correspondence analysis in Figure 7. Following the reviewer's suggestion, we have now included an additional representation of our correspondence analysis between prenatal cell types and the Neftel et al. 2019 GBM dataset malignant cells, using a score-based approach (new Fig. 7d). Specifically, we used the top 50 marker genes within each prenatal population as a module (Supplementary Table 6) and calculated the average enrichment for all prenatal signatures in each GBM malignant cell state (AC, MES1/2, OPC, and NPC1/2, assigned by Neftel et al). This analysis, visualized as a heatmap in new Fig. 7d, displays the average enrichment score for all fetal cell types for each GBM malignant cell state, therefore removing hard assignments and enabling a softer categorization. This broader assessment of developmental correspondence highlights indeed the preferential enrichment of gIPC-A in AC-like GBM cells and gIPC-O in NPC1-like GBM cells. Considering that our new soft-assignment results correspond closely to our previous hard assignment analysis, we prefer to include the pie charts showing the proportion of top predicted cell types in Fig. 7f, in order to highlight further the enrichment of the newly defined gIPC-A and gIPC-O developmental signatures in the context of disease. We would be happy to include this analysis as supplementary information, if preferred by the reviewer. To make this representation less definitive, we have also removed the calculated percentage numbers from Fig. 7f and the text (page 16).